# Analysis of Posture Parameters in Patients with Idiopathic Scoliosis with the Use of 3D Ultrasound Diagnostics—Preliminary Results

**DOI:** 10.3390/ijerph19084750

**Published:** 2022-04-14

**Authors:** Sandra Trzcińska, Michał Kuszewski, Kamil Koszela

**Affiliations:** 1Department of Physiotherapy, College of Rehabilitation in Warsaw, 01-234 Warsaw, Poland; sandra-trzcinska@wp.pl; 2Institute of Physiotherapy and Health Sciences, Academy of Physical Education, 40-065 Katowice, Poland; m.kuszewski@awf.katowice.pl; 3Neuroorthopedics and Neurology Clinic and Polyclinic, National Institute of Geriatrics, Rheumatology and Rehabilitation, 02-637 Warsaw, Poland

**Keywords:** idiopathic scoliosis, scoliosis, diagnosis of scoliosis, scoliometer, scolioscan, 3D ultrasound diagnostics

## Abstract

(1) Background: Idiopathic scoliosis occurs in 2 to 4% of children between 10 and 16 years of age. Due to the harmfulness of radiological examination, non-invasive devices, including the 3D ultrasound technology and Scolioscan apparatus, are more and more frequently used for postural diagnostics. The aim of the study was to analyze the parameters of posture in patients with idiopathic scoliosis with the use of 3D ultrasound diagnostics. (2) Methods: The study included 20 girls, aged 10 to 16 years, with double-curve idiopathic scoliosis (the value of primary curve ranged from 25–50°), types I and II according to King–Moe classification. On the basis of X-ray scan, the Cobb angle of primary and secondary curves was assessed, the skeletal maturity was evaluated with the Risser test, and the type of scoliosis was determined. The girls participated in a 3-week rehabilitation program. The examinations were performed before and after therapy. A scoliometer was used for measurements. Each of the participants underwent individual therapy. The three-plane approach to asymmetric exercises was based largely on positions that included primary curve correction with hypercorrection of the secondary curve. (3) Results: After the therapy, values of trunk rotation angles and the angle of scoliotic curvature of secondary curve were significantly lower than before the therapy, except for the value of the primary curve angle. The parameters measured by X-ray were significantly and positively related to the results obtained with the scoliometer and the scolioscan. (4) Conclusions: The application of therapy that takes into account summing parameters may prevent the deterioration of the secondary curve when treating patients with idiopathic scoliosis. The observed correlations between the parameters of the radiological examination, the scoliometer, and the scolioscan prove the possibility of their interchangeable application in the assessment of effects of the therapy. Three-dimensional ultrasound diagnostics may become an alternative to radiological examination in assessing the treatment effects of patients with idiopathic scoliosis.

## 1. Introduction

Scoliosis is a three-dimensional deviation of the spine axis exceeding 10° Cobb angle in the radiographic image [1,2]. Idiopathic scoliosis occurs in 2 to 4% of children between 10 and 16 years of age [3,4]. The main risk factors for progression are the large curve magnitude, skeletal immaturity, and female gender [5]. Mild scoliosis is usually asymptomatic; however, it may contribute to back pain of musculoskeletal origin [6]. Therefore, its early detection is considered a key procedure in the treatment process, which is difficult due to its unknown etiology [7]. Conservative treatment is aimed at reducing the number of surgeries by preventing the progression of scoliotic deformity [8]. The three-dimensional nature of scoliosis complicates the therapy due to its influence on many parameters of posture assessment. The proved harmfulness of radiation used to assess the spine in terms of scoliosis leads to the limitation of this examination in order to evaluate the effects of the therapy [9]. Therefore, non-invasive apparatuses are more and more frequently used, the harmlessness of which facilitates the assessment of posture in every dimension. The performed therapy for the primary scoliosis curve affects other parameters of the posture examination, and it may cause negative effects in compensatory curves, causing their extension. The use of summing parameters in the scoliometer examination is a quick test of the assessment of the effect of therapy on the overall rotation of the spine and the behavior of the primary curve correction against secondary ones [10].

The application of modern computer diagnostics to assess body posture is recommended but is less used due to the high cost of the apparatus. Nevertheless, examining the patient’s whole posture can prevent a situation in which one of the parameters improves at the expense of others. In recent years, 3D ultrasound technology and the scolioscan apparatus were used for non-invasive and real-time spine examinations [11].

The aim of the study was to analyze the parameters of posture in patients with idiopathic scoliosis with the use of 3D ultrasound diagnostics. The effect on secondary arches in scoliotic deformation was assessed.

## 2. Materials and Methods

### 2.1. Study Population

The study included 20 girls, aged 10 to 16 years, with double-curve idiopathic scoliosis (the value of primary curve ranged from 25° to 50°), Types I and II, according to King–Moe classification. The study was approved by the Bioethics Committee for Scientific Research at Jerzy Kukuczka Academy of Physical Education in Katowice, number 6/2020.

Inclusion criteria:-Current X-ray scan of the pelvic girdle, diagnosed double-curve idiopathic scoliosis of Types I and II according to the King–Moe classification, with the Cobb angle between 25 and 50 degrees of primary scoliosis;-Girls aged 10–16 years;-Unfinished ossification with Risser sign < 5;-No contraindications to the therapy from other systems;-Consent to examination procedures.

Exclusion criteria:-Scoliosis of other than idiopathic origin;-Coexisting diseases of other organs that prevent participation in the rehabilitation program;-Lack of consent of the patient and the guardian to examinations and participation in the program.

### 2.2. Study Protocol

On each radiograph, the following were assessed: Cobb deformation angles of both primary and secondary curve, Risser’s skeletal maturity test, and a specific type of scoliosis. The girls participated in 3-week rehabilitation program. Each patient exercised once a day for one hour from Monday to Friday. Exercises were conducted individually. Three-plane treatment, derotation, shifting, asymmetric stretching, and central stabilization exercises were used. The patients were additionally equipped with a Boston orthopedic brace, which was worn 22–23 h a day. Each of the examined patients could withdraw from the therapy at any time.

Each participant was examined with a Bunnell scoliometer and a Scolioscan apparatus (Telefield Medical Imaging Limited, Hongkong, China). The examinations were performed before and after the rehabilitation program.

Bunnell’s scoliometer is a simple device used to measure the rotation of the trunk; it is characterized by high sensitivity and repeatability [12]. The examination was conducted with the Adams test. The areas of the spine with the highest rotation value were selected, in both primary and secondary scoliosis. The trunk rotation angle (ATR) and the parameter of the sum of two rotations (SDR) were assessed using a scoliometer.

The measurement of the angle of trunk rotation (ATR) was performed at the apex of the scoliosis in the thoracic and lumbar sections with the determination of the primary curve (ATR P) and the secondary curve (ATR W).

The measurement of the sum of two rotations (SDR) consisted of summing two values of the angle of trunk rotation: at the apex of the thoracic curve and at the apex of the lumbar spine.

The value of rotation measured with the accuracy of 1° of left-sided and right-sided curves was summed up as positive value. Simplification of the measurement provided a quick overview of the behavior of both sections—primary and secondary (thoracic and lumbar)—and was an assessment of the global value of trunk rotation in double-curve scoliosis [13].

Then, each of the girls was examined with the Scolioscan apparatus, which uses 3D ultrasound. The severity of the scoliotic deformity was assessed by displaying high-resolution ultrasound images in real time. The examination was carried out in the patient’s habitual position, which was stabilized with movable boards and supporters. Then, an ultrasound gel was applied to the patient’s spine, and the scan area was determined. The advancement of the probe began below the spinous process of the 5th lumbar vertebra and ended automatically after crossing the upper limit of the scan—the first thoracic vertebrae. During scanning, the ultrasound probe was moved from the bottom to the top of the back, covering the whole area of the spine. The test report was saved in PDF format. The process of spine scanning takes about 30–40 s, and creating an image takes about 2 min. The total duration of the examination of the patient was 10–15 min.

The scolioscan test was used to perform angular measurements of scoliosis in the form of the Spinous Process Angle (SPA) parameter. This angle is measured from the centerline of the projected image, which represents the shadow of the spinous processes of the spine vertebrae. It was possible to calculate the angle of the scoliotic deformity by identifying the most tilted part of the spinous process shadow.

Each participant underwent a therapy consisting of self-correction in functional positions, stabilization of the central thorax, asymmetric exercises, and stretching of contracted muscles. Asymmetric therapy was selected individually according to the scoliotic deformity and the patient’s abilities. The three-plane approach to asymmetric exercises was based mainly on positions that included the correction of the primary curve with securing the secondary curve and hypercorrection of the secondary so that it would not be extended during therapy.

### 2.3. Data Analysis

In order to answer the research questions, statistical analyzes were performed with the IBM SPSS Statistics 27 (Armonk, NY, USA). It was used to analyze basic descriptive statistics with Shapiro–Wilk test, Student’s *t*-test for dependent samples and Pearson’s *r* correlation analysis. The level of significance was considered to be α = 0.05.

## 3. Results

### 3.1. Participant Characteristics

On the basis of the analysis of current X-ray scan and the inclusion and exclusion criteria, 20 girls aged 10 to 16 years (mean 14.05 ± 1.64) were enrolled in the study. They were diagnosed with double-curve idiopathic scoliosis (the value of the primary curve was in the range of 25–50°) of Types I and II according to the King–Moe classification.

The patients weighed 36 to 73 kg (mean 52.50 ± 9.45) and measured 143 to 174 cm (mean 163.40 ± 7.16). The average BMI was 19.58 ± 2.69. The mean Cobb angle of the primary curve (°) Cobb P was 32.45 ± 6.53 in the subjects, and the secondary curve (°) Cobb W was 21.85 ± 7.91.

In the Risser test, girls obtained a grade of 0 to 4, and the mean grade was 2.50 ± 1.67. Most of the patients (85.0%) presented Type II scoliosis according to the King–Moe classification. Detailed results are presented in Table 1 and Table 2.

### 3.2. The Shapiro–Wilk Test

In the first step of the analysis, the distributions of quantitative variables were checked. For this purpose, simple descriptive statistics were calculated together with the Shapiro–Wilk test examining the normality of the distribution. The results of the analysis are presented in Table 3.

The results of the Shapiro–Wilk test appeared to be statistically significant only for the Cobb angle of the primary curve (Cobb P) before the therapy and for the trunk rotation angle of the primary curve (ATR P) after the therapy. This means that the distributions of these variables differ from normal distribution. However, in both cases, the skewness did not exceed the absolute value of two, which indicates a slight asymmetry of the distributions. Therefore, parametric tests were used to verify the hypotheses.

### 3.3. Comparison of the Values of Rotation and Curvature Angles

In the next step, it was checked whether the applied therapy influenced the values of the trunk rotation angles of the primary (ATR P) and secondary curves (ATR W), the sum of two rotations (SDR), and the scoliotic curvature angles of the primary (SPA P) and secondary curves (SPA W). For this purpose, the Student’s *t*-test was performed for dependent samples, comparing parameters before and after the therapy (Table 4, Figure 1).

After the therapy, the values of the trunk rotation angles (primary and secondary—ATR P, ATR W, and SDR) and the angle of scoliotic curvature of the secondary curve (SPA W) were significantly lower than before the therapy. The analysis showed statistically significant differences in all parameters, except for the value of the angle of scoliotic curvature of the primary curve (SPA W).

### 3.4. Correlations between Parameters Measured with X-ray, Scoliometer, and Scolioscan

In the last step, the hypotheses regarding the correlation of measurements made with the use of X-ray, scoliometer, and scolioscan were verified. For this purpose, Pearson’s *r* correlation analysis was performed. The parameters were compared both before and after therapy. The results are presented in Table 5.

The analysis showed six statistically significant correlations. The parameters measured by X-ray were significantly and positively related to the results obtained with the scoliometer and the scolioscan. The correlation between the Cobb angle of the secondary curve (Cobb W) and the trunk rotation angle of the secondary curve (ATR W) was moderate, and the remaining ones were strong.

The results for the scoliometer and the scolioscan were related to each other in terms of the value of the angle of trunk rotation of the secondary curve (ATR W) and the angle of the scoliotic curvature of the secondary curve (SPA W), both before and after the therapy. Both correlations were positive and moderately strong. On the other hand, the trunk rotation angle of the primary curve (ATR P) measured with a scoliometer was not statistically significantly correlated with the analogous measurement (angle of scoliotic curvature of primary curve (SPA P)) performed with the scolioscan. Correlations for these measurements were statistically insignificant both before and after therapy.

## 4. Discussion

In 2021, studies were conducted on the effectiveness of the FEF and FITS methods in the treatment of idiopathic scoliosis. These studies have shown that both methods significantly improved the rotation of the trunk, in both the primary and secondary curves, but after their summation (SDR parameter), the FED method was statistically more effective [14]. The above results do not undermine the effectiveness of the FITS method in the treatment of scoliosis but indicate an existing problem concerning the extension of compensatory curves during the correction of the primary curve. The FITS method is recognized by SOSORT and widely used in Poland as the leading method in the treatment of patients with idiopathic scoliosis [15]. The authors of the FITS method themselves observed such a phenomenon, explaining it as the development of the compensation mechanism, and they expected confirmation of their own observations regarding the increase in the rotation of the trunk within the compensatory curves during the therapy [16].

Perhaps at this point a modification of the approach to the treatment of scoliosis would be worth considering, not so much for changing the method of therapy but for controlling of the secondary curve by introducing an appropriate procedure that is based on the diagnosis of the whole posture. Usually studies on the effectiveness of different methods concern separate assessment of primary and secondary curves and less frequently they assess the global, overall rotation of the trunk, which is responsible for the reaction of the whole spine to the therapy. Introduction of summing parameters is not yet a standard diagnostic procedure, but reports on their use appear more and more frequently and prove their appropriateness in assessing the impact of treatment of scoliotic deformity [17].

The results of the conducted studies have demonstrated that after the applied therapy based on summing parameters, there was a significant decrease in both parameters of the position of primary (ATR P) and secondary (ATR W) curves. The SDR summation parameter, which is responsible for the overall rotation of the spine in double-curve scoliosis, also improved.

The therapy applied in this study consisted of reversing the compensation mechanisms that cause the expansion of the compensatory curves in idiopathic scoliosis. Compensatory treatment is therefore based on the use of summing parameters that assess the effect of therapy on the global rotation of the spine and both curves as well as on the use of asymmetric, three-plane exercises, which have the same corrective effect on primary and secondary curves. The overall assessment of posture is associated with the introduction of modern diagnostics, which in a non-invasive and harmless manner would prevent situations in which the improvement of one of the scoliosis parameters causes the deterioration of others.

Much has been said about the harmfulness of X-ray exposure, but nevertheless it is still the gold standard in the diagnosis of patients with idiopathic scoliosis [18]. This examination is irreplaceable when it comes to diagnosing this deformity and assessing its progression; however, in recent years, non-invasive and non-harmful devices have been used more and more often to assess the effectiveness of therapy, which allow the examination to be carried out at any time.

Recently, a new apparatus has appeared on the medical market, which is the first in the world to study scoliosis using 3D ultrasound technology. Owing to its harmlessness, it can be used many times at any stage of therapy: it can be used for screening, diagnosis, progression monitoring, and evaluation of treatment effectiveness. It creates a three-dimensional model of the spine by registering its real shape [19,20,21,22,23]. Therefore, it may become an alternative to traditional radiological examination as it correlates with the Cobb angle [19]. The occurring correlations between the parameters of posture examination performed with various research tools indicate that they can be used to monitor the treatment of patients with idiopathic scoliosis.

The conducted studies have demonstrated the existence of statistically significant correlations between the parameters measured with X-ray, scoliometer, and scolioscan. The smallest correlation was observed between the scolioscan and the scoliometer in terms of primary curve measurements. Similar observations concern the therapeutic effect of the SPA P parameter. Scolioscan did not show any significant improvement in this parameter after the therapy. This is the reason for further considerations of the correlation between the scoliometer—a simple and useful tool for measuring the rotation of the trunk in the transverse plane and the scolioscan giving a real image of the spine. As of today, the medical world focuses mainly on the correlation between scolioscan and radiological parameters, but it is possible there will appear other devices to monitor the effects and evaluate the use of the scolioscan for screening patients with idiopathic scoliosis.

Three-dimension ultrasound diagnostics offers new diagnostic possibilities; however, it requires further analyses supplemented with long-term follow-up in a larger group of patients.

## 5. Conclusions

The application of therapy that takes into account summing parameters may prevent the deterioration of the secondary curve when treating patients with idiopathic scoliosis.The observed correlations between the parameters of the radiological examination, the scoliometer and the scolioscan, prove the possibility of their interchangeable application in the assessment of effects of the therapy.Three-dimensional ultrasound diagnostics may become an alternative to radiological examination in assessing the treatment effects of patients with idiopathic scoliosis.

## Figures and Tables

**Figure 1 ijerph-19-04750-f001:**
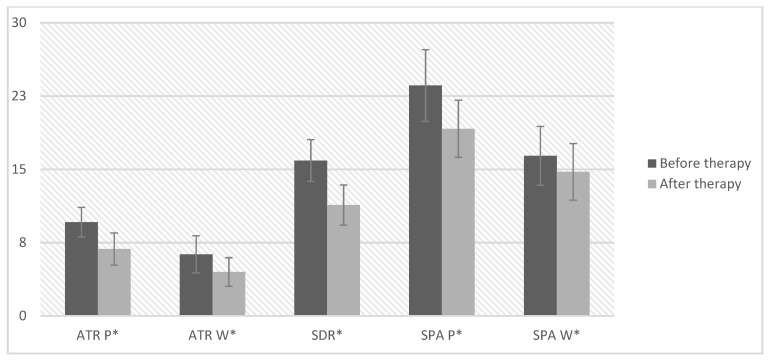
Mean values with 95% confidence intervals before and after therapy for measurements of angles of trunk rotation of primary and secondary curves as well as angles of scoliotic curvature of primary and secondary curves, *—statistical significance.

**Table 1 ijerph-19-04750-t001:** Participant characteristics (*n* = 20).

	$\bar{x}$ ± *SD*	*Min*	*Max*
**Age (years)**	14.05 ± 1.64	10.00	16.00
**Weight (kg)**	52.50 ± 9.45	36.00	73.00
**Hight (cm)**	163.40 ± 7.16	143.00	174.00
**BMI**	19.58 ± 2.69	15.79	25.86
**Cobb angle of the primary curve (** **°** **) Cobb P**	32.45 ± 6.53	25.00	50.00
**Cobb angle of the secondary curve (** **°** **) Cobb W**	21.85 ± 7.91	11.00	41.00
**Risser test (points)**	2.50 ± 1.67	0.00	4.00

*n*—number, $\bar{x}$—mean, *SD*—standard deviation, *Min*—the lowest value, *Max*—the highest value, *BMI*—body mass index.

**Table 2 ijerph-19-04750-t002:** Type of scoliosis based on King–Moe Classification.

King–Moe Classification	*n*	% of Total
Type I	3	15.0
Type II	17	85.0
Total	20	100.0

*n*—number.

**Table 3 ijerph-19-04750-t003:** Simple descriptive statistics of the studied variables together with the Shapiro–Wilk normality test.

	$\bar{x}$ ± *SD*	*Me*	*Kurt*	*Min*	*Max*	*W*	*p*
**Before therapy (*n* = 20)**							
**Cobb angle of the primary curve (** **°** **) Cobb P**	32.45 ± 6.53	31.50	2.01	25.00	50.00	0.87	0.011 *
**Cobb angle of the secondary curve (** **°** **) Cobb W**	21.85 ± 7.91	22.50	0.45	11.00	41.00	0.95	0.343
**The trunk rotation angle primary curve (** **°** **) ATR P**	9.6 ± 3.45	9.00	1.61	5.00	19.00	0.92	0.121
**The trunk rotation angle secondary curve (** **°** **) ATR W**	6.3 ± 4.34	6.00	−0.19	0.00	16.00	0.95	0.378
**Sum of two rotations (** **°** **) SDR**	15.9 ± 4.88	15.00	−0.60	9.00	26.00	0.95	0.316
**Angle of scoliotic curvature of the primary curve (** **°** **) SPA P**	23.58 ± 8.37	20.45	−1.14	12.00	38.60	0.93	0.158
**Angle of scoliotic curvature of the secondary curve (** **°** **) SPA W**	16.39 ± 6.86	17.85	1.97	0.00	33.50	0.93	0.177
**After therapy (*n* = 20)**							
**The trunk rotation angle primary curve (** **°** **) ATR P**	6.85 ± 3.76	6.00	2.96	2.00	18.00	0.86	0.009 *
**The trunk rotation angle secondary curve (** **°** **) ATR W**	4.5 ± 3.35	4.00	−0.73	0.00	11.00	0.94	0.212
**Sum of two rotations(** **°** **) SDR**	11.35 ± 4.69	11.50	0.04	5.00	22.00	0.92	0.113
**Angle of scoliotic curvature of the primary curve (** **°** **) SPA P**	19.15 ± 6.67	19.10	−1.14	9.20	30.40	0.95	0.371
**Angle of scoliotic curvature of the secondary curve (** **°** **) SPA W**	14.74 ± 6.62	15.80	−0.28	0.00	25.00	0.97	0.766

*n*—number, $\bar{x}$—mean, *SD*—standard deviation, *Me*—median, *Kurt*—kurtosis, *Min*—the lowest value, *Max*—the highest value, *W*—Shapiro–Wilk test, *p*—level of significance, *—statistical significance.

**Table 4 ijerph-19-04750-t004:** Results of Student’s *t*-test for dependent samples—comparison of trunk rotation angles of primary and secondary curves and angles of scoliotic curvature of primary and secondary curves, before and after therapy.

	Before Therapy (*n* = 20)	After Therapy (*n* = 20)	Average Difference	*t*	*p*	95% *CI*		*d* Cohena
	$\bar{x}$ ± *SD*	$\bar{x}$ ± *SD*				** *LL* **	** *UL* **	
**The trunk rotation angle primary curve** **(** **°** **) ATR P**	9.6 ± 3.45	6.85 ± 3.76	2.75	8.10	<0.001 *	2.04	3.46	1.81
**The trunk rotation angle secondary curve** **(** **°** **) ATR W**	6.3 ± 4.34	4.5 ± 3.35	1.80	3.45	0.002 *	0.71	2.89	0.77
**Sum of two rotations (** **°) SDR**	15.9 ± 4.88	11.35 ± 4.69	4.55	7.42	<0.001 *	3.27	5.83	1.66
**Angle of scoliotic curvature of the primary curve** **(** **°** **) SPA P**	23.58 ± 8.37	19.15 ± 6.67	4.43	3.64	0.001 *	1.88	6.98	0.81
**Angle of scoliotic curvature of the secondary curve** **(** **°** **) SPA W**	16.39 ± 6.86	14.74 ± 6.62	1.65	2.00	0.059 *	−0.07	3.37	0.45

*n*—number, $\bar{x}$—mean, *SD*—standard deviation, *t*—*t*-test, *p*—level of significance, 95% *CI*—Confidence Interval, *LL*—Lower Limit, *UL*—Upper Limit, *d Cohena*—effect size, *—statistical significance.

**Table 5 ijerph-19-04750-t005:** Analysis of the correlation of measurements made in patients with the use of X-ray, scoliometer and scolioscan, before and after therapy.

Compared Methods	Compared Variables			Before Therapy	After Therapy
**X-ray—Skoliometr**	Cobb angle of the primary curve (°) Cobb P	The trunk rotation angle primary curve (°) ATR P	*r* Pearson	0.54	-
			*p*	0.014 *	-
	Cobb angle of the secondary curve (°) Cobb W	The trunk rotation angle secondary curve (°) ATR W	*r* Pearson	0.48	-
			*p*	0.031 *	-
**X-ray—Skolioscan**	Cobb angle of the primary curve (°) Cobb P	Angle of scoliotic curvature of the primary curve (°) SPA P	*r* Pearson	0.7	-
			*p*	0.001 *	-
	Cobb angle of the secondary curve (°) Cobb W	Angle of scoliotic curvature of the secondary curve (°) SPA W	*r* Pearson	0.62	-
			*p*	0.003 *	-
**Skoliometr—Scolioscan**	The trunk rotation angle primary curve (°) ATR P	Angle of scoliotic curvature of the primary curve (°) SPA P	*r* Pearson	0.44	0.35
			*p*	0.054 *	0.132
	The trunk rotation angle secondary curve (°) ATR W	Angle of scoliotic curvature of the secondary curve (°) SPA W	*r* Pearson	0.47	0.44
			*p*	0.036 *	0.050 *

*p*—level of significance, *—statistical significance.

## Data Availability

The datasets analyzed during the current study are available from the corresponding author upon reasonable request.

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
