# Peer review of "Analysis of Posture Parameters in Patients with Idiopathic Scoliosis with the Use of 3D Ultrasound Diagnostics—Preliminary Results"

_ijerph, 2022, doi:10.3390/ijerph19084750_

Round 1
Reviewer 1 Report
Thank you for the privilege of reviewing this article. The authors describe in their article an analysis of posture parameters 20 children aged 10 to 16 years of age suffering from idiopathic scoliosis who underwent 3D ultrasound diagnostics for monitoring of conservative treatment. The article is well written and the statistics appear correctly performed to me.
I have a few questions and recommendations:
Did you apply a sample size calculation?
What was the study interval?
Please include a chapter describing limitations and strengths of your investigation (at the end of discussion section)
Please correct some words and phrases:
Line 60: replace: “are” by “were”.
Line 89-90: replace: “24 hours a day, except for therapy and personal hygiene (22-23 hours a day).” by “22-23 hours a day.”.
Line 146, 151: replace: “subjects” by “patients”. (please be humble and use patients instead of subjects)
Line 154: replace: “Participant characteristics” by “Participant characteristics (n = 20)”.
Line 154: delete: first column (n).
Line 195-196: delete sentence: “The observed … measurements”.
Line 209: replace “showed the occurrence of” by “showed”.
Line 237: replace “considering. Not” by “considering, not”.
Author Response
Dear Reviewer,
Thank you very much for your valuable comments.
Of course, we used sample size calculations. The value was 0.91.
The patients had a break on Sunday, the rehabilitation process was conducted from Monday to Friday.
At the end of discussion section we added limitations and strengths of our investigation.
We changed as you suggest in line: 60, 89-90, 146, 160, 154, 195-196, 209, 237.
Reviewer 2 Report
The manuscript is relevant and has signs of novelty. The author presents a fresh approach to research posture parameters in patients with idiopathy.
This manuscript is recommended for publication after major corrections.
Remarks:
The aim of the study is to analyze the parameters of posture. The authors but does not specify why this research and analysis has been performed. I suggest extending and specifying the aim of the study.
I suggest specifying the methodology with the specifications of the test equipment were used.
Information on the ethical aspects of the study is required.
It is not necessary to provide a tool for statistical calculations, it is sufficient to indicate the methods and criteria of the applied statistics.
In Table 1, the information on the number of subjects is redundant.
Table 3 requires an explanation.
It is necessary to extend the findings to include research data to support the claims.
Author Response
Dear Reviewer,
Thank you very much for your valuable comments.
The aim of the study is to analyze the parameters of posture. The authors but does not specify why this research and analysis has been performed. I suggest extending and specifying the aim of the study.
We have made a correction.
The aim of the study was to analyze the parameters of posture in patients with idiopathic scoliosis with the use of 3D ultrasound diagnostics. The effect on secondary arches in scoliotic deformation was assessed.
Information on the ethical aspects of the study is required.
We added more information to "Institutional Review Board Statement:"
This study was conducted according to the guidelines of the Declaration of Helsinki, and approved by the Bioethics Committee for Scientific Research at Jerzy Kukuczka Academy of Physical Education in Katowice, number 6/2020.
In Table 1, the information on the number of subjects is redundant.
We changed it.
Table 3 requires an explanation.
We have completed the legend of table No 3 and 4.
Reviewer 3 Report
The conclusions want to be more detailed and correlated with the results. Correcting small grammar mistakes.
Author Response
Dear Reviewer,
Thank you very much for your valuable comments.
We modified the conclusion No.3.
3. 3D ultrasound diagnostics may become an alternative to radiological examination in evaluating the effects of treatment with idiopathic scoliosis.
Reviewer 4 Report
Dear authors, the idea of using 3D-ultrasound diagnostics is very interesting.However, the study needs some substantial changes.
The populations is not homogeneous to the type of treatment.
According to the literature scoliosis <20 ° require observation and / or physiotherapy;
scoliosis <25 ° require orthopedic brace;
> 40 ° requires surgery.
Therefore all the data collected are falsified by unsuitable treatment.
The rehabilitation treatment is not described: there is not number of sessions,
duration of sessions, type of exercises in detail.
It is not specified with or without a corset
It is necessary to review all the statistical analysis there are many mistakes.
The conclusion is not supported by the results. It cannot be said that the 3D ultrasound method can replace x-ray (preliminary study) In the literature, x-ray is not used to monitor therapy
Author Response
Dear Reviewer,
Thank you very much for your valuable comments.
It is our mistake. It should be 25 degrees. The smallest tilt value is 25, it is shown in table 1. We have corrected it in the text.
We added to the text (2.2. Study Protocol).
"Each patient was exercised once a day for one hour from Monday to Friday. Exercises conducted individually. Three-plane treatment, derotation, shifting, asymmetric stretching, central stabilization exercises were used."
The statistical analysis has been checked by a statistician.
Conclusion No 3 has been revised.
Reviewer 5 Report
I have the following suggestions for the improvement of the manuscript:
Please Try to mention data significance in a clearer way. Tables showing p values are difficult to focus on for the readers. Better include '*' to clearly focus on the significant values.
Tables or parts of tables could be replaced with graphs (if possible) for better illustration.
To mention values, I think 14,05 ± 1,64 10,00 16,00 could be replaced by 14.05 ± 1.64 10.00 16.00.
Please show the significance sign on the Figure.
Author Response
Dear Reviewer,
Thank you very much for your valuable comments.
Please Try to mention data significance in a clearer way. Tables showing p values are difficult to focus on for the readers. Better include '*' to clearly focus on the significant values.
We changed it.
Tables or parts of tables could be replaced with graphs (if possible) for better illustration.
If necessary, we will try but it will be difficult.
To mention values, I think 14,05 ± 1,64 10,00 16,00 could be replaced by 14.05 ± 1.64 10.00 16.00.
We changed it.
Please show the significance sign on the Figure.
We did it.
Round 2
Reviewer 2 Report
Thank the authors for constructive correction of the manuscript.
The resubmitted manuscript can be published.
Sincerely.
Reviewer 4 Report
Add in the introduction to the line 3-7 of information on epidemiology and risk factors, I recommend the following article:1) Scaturro D, Costantino C, Terrana P, Vitagliani F, Falco V, Cuntrera D, Sannasardo CE, Vitale F, Letizia Mauro G. Risk Factors, Lifestyle and Prevention among Adolescents with Idiopathic Juvenile Scoliosis: A Cross Sectional Study in Eleven First-Grade Secondary Schools of Palermo Province, Italy. Int J Environ Res Public Health. 2021 Nov 24;18(23):12335. doi: 10.3390/ijerph182312335. PMID: 34886069; PMCID: PMC8656498.
2) Scaturro D, de Sire A, Terrana P, Costantino C, Lauricella L, Sannasardo CE, Vitale F, Mauro GL. Adolescent idiopathic scoliosis screening: Could a school-based assessment protocol be useful for an early diagnosis? J Back Musculoskelet Rehabil. 2021;34(2):301-306. doi: 10.3233/BMR-200215. PMID: 33285626.